# Predicting Scene Parsing and Motion Dynamics in the Future

**Xiaojie Jin**[1], **Huaxin Xiao**[2], **Xiaohui Shen**[3], **Jimei Yang**[3], **Zhe Lin**[3]
**Yunpeng Chen**[2], **Zequn Jie**[4], **Jiashi Feng**[2], **Shuicheng Yan**[5,2]
[1]NUS Graduate School for Integrative Science and Engineering (NGS), NUS
[2]Department of ECE, NUS    [3]Adobe Research    [4]Tencent AI Lab    [5]Qihoo 360 AI Institute

## Abstract

The ability of predicting the future is important for intelligent systems, *e.g.* autonomous vehicles and robots to plan early and make decisions accordingly. Future scene parsing and optical flow estimation are two key tasks that help agents better understand their environments as the former provides dense semantic information, *i.e.* what objects will be present and where they will appear, while the latter provides dense motion information, *i.e.* how the objects will move. In this paper, we propose a novel model to simultaneously predict scene parsing and optical flow in unobserved future video frames. To our best knowledge, this is the first attempt in jointly predicting scene parsing and motion dynamics. In particular, scene parsing enables structured motion prediction by decomposing optical flow into different groups while optical flow estimation brings reliable pixel-wise correspondence to scene parsing. By exploiting this mutually beneficial relationship, our model shows significantly better parsing and motion prediction results when compared to well-established baselines and individual prediction models on the large-scale Cityscapes dataset. In addition, we also demonstrate that our model can be used to predict the steering angle of the vehicles, which further verifies the ability of our model to learn latent representations of scene dynamics.

## 1   Introduction

Future prediction is an important problem for artificial intelligence. To enable intelligent systems like autonomous vehicles and robots to react to their environments, it is necessary to endow them with the ability of predicting what will happen in the near future and plan accordingly, which still remains an open challenge for modern artificial vision systems.

In a practical visual navigation system, scene parsing and dense motion estimation are two essential components for understanding the scene environment. The former provides pixel-wise prediction of semantic categories (thus the system understands what and where the objects are) and the latter describes dense motion trajectories (thus the system learns how the objects move). The visual system becomes "smarter" by leveraging the prediction of these two types of information, *e.g.* predicting how the car coming from the opposite direction moves to plan the path ahead of time and predict/control the steering angle of the vehicle. Despite numerous models have been proposed on scene parsing [4, 7, 17, 26, 28, 30, 15] and motion estimation [2, 9, 21], most of them focus on processing observed images, rather than predicting in unobserved future scenes. Recently, a few works [22, 16, 3] explore how to anticipate the scene parsing or motion dynamics, but they all tackle these two tasks separately and fail to utilize the benefits that one task brings to the other.

In this paper, we try to close this research gap by presenting a novel model for jointly predicting scene parsing and motion dynamics (in terms of the dense optical flow) for future frames. More importantly, we leverage one task as the auxiliary of the other in a mutually boosting way. See Figure 1 for

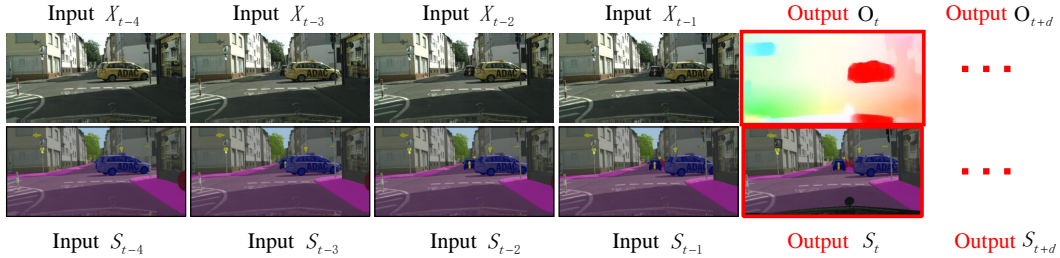

Figure 1: Our task. The proposed model jointly predicts scene parsing and optical flow in the future. **Top**: Future flow (highlighted in red) anticipated using preceding frames. **Bottom**: Future scene parsing (highlighted in red) anticipated using preceding scene parsing results. We use the flow field color coding from [2].

an illustration of our task. For the task of predictive scene parsing, we use the discriminative and temporally consistent features learned in motion prediction to produce parsing prediction with more fine details. For the motion prediction task, we utilize the semantic segmentations produced by predictive parsing to separately estimate motion for pixels with different categories. In order to perform the results for multiple time steps, we take the predictions as input and iterate the model to predict subsequent frames. The proposed model has a generic framework which is agnostic to backbone deep networks and can be conveniently trained in an end-to-end manner.

Taking Cityscapes [5] as testbed, we conduct extensive experiments to verify the effectiveness of our model in future prediction. Our model significantly improves mIoU of parsing predictions and reduces the endpoint error (EPE) of flow predictions compared to strongly competitive baselines including a warping method based on optical flow, standalone parsing prediction or flow prediction and other state-of-the-arts methods [22]. We also present how to predict steering angles using the proposed model.

## 2 Related work

For the general field of classic flow (motion) estimation and image semantic segmentation, which is out of this paper's scope, we refer the readers to comprehensive review articles [2, 10]. Below we mainly review existing works that focus on predictive tasks.

**Flow and scene parsing prediction**   The research on predictive scene parsing or motion prediction is still relatively under-explored. All existing works in this direction tackle the parsing prediction and flow prediction as independent tasks. With regards to motion prediction, Luo *et al.* [19] employed a convolutional LSTM architecture to predict sequences of 3D optical flow. Walker *et al.* [35] made long-term motion and appearance prediction via a transition and context model. [31] trained CNN for predicting motion of handwritten characters in a synthetic dataset. [36] predicted future optical flow given a static image. Different from above works, our model not only predicts the flow but also scene parsing at the same time, which definitely provides richer information to visual systems.

There are also only a handful number of works exploring the prediction of scene parsing in future frames. Jin *et al.* [16] trained a deep model to predict the segmentations of the next frame from preceding input frames, which is shown to be beneficial for still-image parsing task. Based on the network proposed in [20], Natalia *et al.* [22] predicted longer-term parsing maps for future frames using the preceding frames' parsing maps. Different from [22], we simultaneously predict optical flows for future frames. Benefited from the discriminative local features learned from flow prediction, the model produces more accurate parsing results. Another related work to ours is [24] which employed an RNN to predict the optical flow and used the flow to warp preceding segmentations. Rather than simply producing the future parsing map through warping, our model predicts flow and scene parsing jointly using learning methods. More importantly, we leverage the benefit that each task brings to the other to produce better results for both flow prediction and parsing prediction.

**Predictive learning**   While there are few works specifically on predictive scene parsing or dense motion prediction, learning to prediction in general has received a significant attention from the

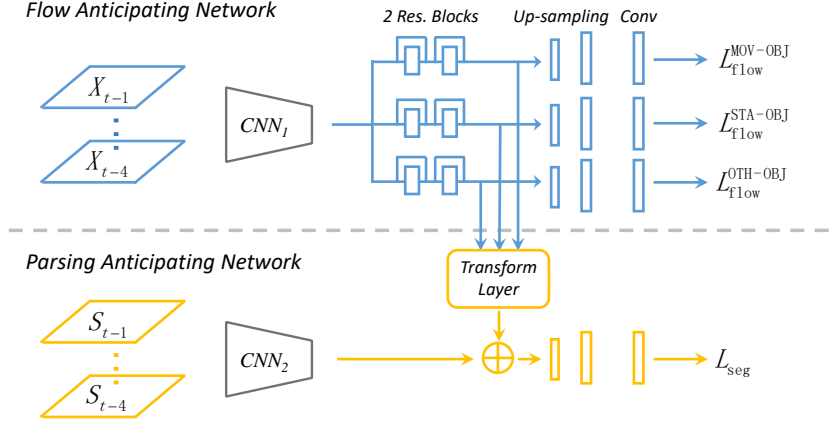

Figure 2: The framework of our model for predicting future scene parsing and optical flow for one time step ahead. Our model is motivated by the assumption that flow and parsing prediction are mutually beneficial. We design the architecture to promote such mutual benefits. The model consists of two module networks, *i.e.* the flow anticipating network (blue) which takes preceding frames: $X_{t-4:t-1}$ as input and predicts future flow and the parsing anticipating network (yellow) which takes the preceding parsing results: $S_{t-4:t-1}$ as input and predicts future scene parsing. By providing pixel-level class information (*i.e.* $S_{t-1}$), the parsing anticipating network benefits the flow anticipating network to enable the latter to semantically distinguish different pixels (*i.e.* moving/static/other objects) and predict their flows more accurately in the corresponding branch. Through the transform layer, the discriminative local features learned by the flow anticipating network are combined with the parsing anticipating network to facilitate parsing over small objects and avoid over-smooth in parsing predictions. When predicting multiple time-steps ahead, the prediction of the parsing network in a time-step is used as the input in the next time-step.

research community in recent years. Research in this area has explored different aspects of this problem. [37] focused on predicting the trajectory of objects given input image. [13] predicted the action class in the future frames. Generative adversarial networks (GAN) are firstly introduced in [11] to generate natural images from random noise, and have been widely used in many fields including image synthesis [11], future prediction [18, 20, 34, 36, 32, 33] and semantic inpainting [23]. Different from above methods, our model explores a new predictive task, *i.e.* predicting the scene parsing and motion dynamics in the future simultaneously.

**Multi-task learning** Multi-task learning [1, 6] aims to solve multiple tasks jointly by taking advantage of the shared domain knowledge in related tasks. Our work is partially related to multi-task learning in that both the parsing results and motion dynamics are predicted jointly in a single model. However, we note that predicting parsing and motion "in the future" is a novel and challenging task which cannot be straightforwardly tackled by conventional multi-task learning methods. To our best knowledge, our work is the first solution to this challenging task.

## 3 Predicting scene parsing and motion dynamics in the future

In this section, we first propose our model for predicting semantics and motion dynamics one time step ahead, and then extend our model to perform predictions for multiple time steps.

Due to high cost of acquiring dense human annotations of optical flow and scene parsing for natural scene videos, only subset of frames are labeled for scene parsing in the current datasets. Following [22], to circumvent the need for datasets with dense annotations, we train an adapted Res101 model (denoted as Res101-FCN, more details are given in Sec. 4.1) for scene parsing to produce the target semantic segmentations for frames without human annotations. Similarly, to obtain the dense flow map for each frame, we use the output of the state-of-the-art epicflow [25] as our target optical flow. Note that our model is orthogonal to specific flow methods since they are only used to produce the target flow for training the flow anticipating network. Notations used in the following text are as follows. $X_i$ denotes the $i$-th frame of a video and $X_{t-k:t-1}$ denotes the sequence of frames with length $k$ from $X_{t-k}$ to $X_{t-1}$. The semantic segmentation of $X_t$ is denoted as $S_t$, which is the

output of the penultimate layer of Res101-FCN. $S_t$ has the same spatial size as $X_t$ and is a vector of length $C$ at each location, where $C$ is the number of semantic classes. We denote $O_t$ as the pixel-wise optical flow map from $X_{t-1}$ to $X_t$, which is estimated via epicflow [25]. Correspondingly, $\hat{S}_t$ and $\hat{O}_t$ denote the predicted semantic segmentation and optical flow.

## 3.1 Prediction for one time step ahead

**Model overview** The key idea of our approach is to model flow prediction and parsing prediction jointly, which are potentially mutually beneficial. As illustrated in Figure 2, the proposed model consists of two module networks that are trained jointly, *i.e.* the flow anticipating network that takes preceding frames $X_{t-k:t-1}$ as input to output the pixelwise flow prediction for $O_t$ (from $X_{t-1}$ to $X_t$), and the parsing anticipating network that takes the segmentation of preceding frames $S_{t-k:t-1}$ as input to output pixelwise semantic prediction for an unobserved frame $X_t$. The mutual influences of each network on the other are exploited in two aspects. First, the last segmentations $S_{t-1}$ produced by the parsing anticipating network convey pixel-wise class labels, which are used by the flow anticipating network to predict optical flow values for each pixel according to its belonging object group, *e.g.* moving objects or static objects. Second, the parsing anticipating network combines the discriminative local feature learned by the flow anticipating network to produce sharper and more accurate parsing predictions.

Since both parsing prediction and flow prediction are essentially both the dense classification problem, we use the same deep architecture (Res101-FCN) for predicting parsing results and optical flow. Note the Res101-FCN used in this paper can be replaced by any CNNs. We adjust the input/output layers of these two networks according to the different channels of their input/output. The features extracted by feature encoders (CNN$_1$ and CNN$_2$) are spatially enlarged via up-sampling layers and finally fed to a convolutional layer to produce pixel-wise predictions which have the same spatial size as input.

**Flow anticipating network** In videos captured for autonomous driving or navigation, regions with different class labels have different motion patterns. For example, the motion of static objects like *road* is only caused by the motion of the camera while the motion of moving objects is a combination of motions from both the camera and objects themselves. Therefore compared to methods that predict all pixels' optical flow in a single output layer, it would largely reduce the difficulty of feature learning by separately modeling the motion of regions with different classes. Following [29], we assign each class into one of three pre-defined object groups, *i.e.* $\mathcal{G} = \{moving\ objects\ (MOV\text{-}OBJ), static\ objects\ (STA\text{-}OBJ), other\ objects\ (OTH\text{-}OBJ)\}$ in which MOV-OBJ includes pedestrians, truck, *etc.*, STA-OBJ includes sky, road, *etc.*, and OTH-OBJ includes vegetation and buildings, *etc.* which have diverse motion patterns and shapes. We append a small network (consisting of two residual blocks) to the feature encoder (CNN$_1$) for each object group to learn specified motion representations. During training, the loss for each pixel is only generated at the branch that corresponds to the object group to which the pixel belongs. Similarly, in testing, the flow prediction for each pixel is generated by the corresponding branch. The loss function between the model output $\hat{O}_t$ and target output $O_t$ is

$$L_{\text{flow}}(\hat{O}_t, O_t) = \sum_{g \in \mathcal{G}} L_{\text{flow}}^g; \quad L_{\text{flow}}^g = \frac{1}{|N_g|} \sum_{(i,j) \in N_g} \left\| O_t^{i,j} - \hat{O}_t^{i,j} \right\|_2 \tag{1}$$

where $(i, j)$ index the pixel in the region $N_g$.

**Parsing anticipating network** The input of the parsing anticipating network is a sequence of preceding segmentations $S_{t-k:t-1}$. We also explore other input space alternatives, including preceding frames $X_{t-k:t-1}$, and the combination of preceding frames and corresponding segmentations $X_{t-k:t-1}S_{t-k:t-1}$, and we observe that the input $S_{t-k:t-1}$ achieves the best prediction performance. We conjecture it is easier to learn the mapping between variables in the same domain (*i.e.* both are semantic segmentations). However, there are two drawbacks brought by this strategy. Firstly, $S_{t-k:t-1}$ lose the discriminative local features *e.g.* color, texture and shape *etc.*, leading to the missing of small objects in predictions, as illustrated in Figure 3 (see yellow boxes). The flow prediction network may learn such features from the input frames. Secondly, due to the lack of local features in $S_{t-k:t-1}$, it is difficult to learn accurate pixel-wise correspondence in the parsing anticipating

network, which causes the predicted labeling maps to be over-smooth, as shown in Figure 3. The flow prediction network can provide reliable dense pixel-wise correspondence by regressing to the target optical flow. Therefore, we integrate the features learned by the flow anticipating network with the parsing prediction network through a transform layer (a shallow CNN) to improve the quality of predicted labeling maps. Depending on whether human annotations are available, the loss function is defined as

$$L_{\text{seg}}(\hat{S}, S) = \begin{cases} -\sum\limits_{(i,j) \in X_t} \log(\hat{S}_t^{i,j}(c)), & X_t \text{ has human annotation,} \\ L_{\ell_1}(\hat{S}, S) + L_{\text{gdl}}(\hat{S}, S), & \text{otherwise} \end{cases} \quad (2)$$

where $c$ is the ground truth class for the pixel at location $(i, j)$. It is a conventional pixel-wise cross-entropy loss when $X_t$ has human annotations. $L_{\ell_1}$ and $L_{\text{gdl}}$ are $\ell_1$ loss and gradient difference loss [20] which are defined as

$$L_{\ell_1}(\hat{S}, S) = \sum_{(i,j) \in X_t} \left| S_t^{i,j} - \hat{S}_t^{i,j} \right|,$$

$$L_{\text{gdl}} = \sum_{(i,j) \in X_t} \left( \left| |S_t^{i,j} - S_t^{i-1,j}| - |\hat{S}_t^{i,j} - \hat{S}_t^{i-1,j}| \right| + \left| |S_t^{i,j-1} - S_t^{i,j}| - |\hat{S}_t^{i,j-1} - \hat{S}_t^{i,j}| \right| \right).$$

The $\ell_1$ loss encourages predictions to regress to the target values while the gradient difference loss produces large errors in the gradients of the target and predictions.

The reason for using different losses for human and non-human annotated frames in Eq. 2 is that the automatically produced parsing ground-truth (by the pre-trained Res101-FCN) of the latter may contain wrong annotations. The cross-entropy loss using one-hot vectors as labels is sensitive to the wrong annotations. Comparatively, the ground-truth labels used in the combined loss ($L_{\ell_1} + L_{\text{gdl}}$) are inputs of the softmax layer (ref. Sec. 3) which allow for non-zero values in more than one category, thus our model can learn useful information from the correct category even if the annotation is wrong. We find replacing $L_{\ell_1} + L_{\text{gdl}}$ with the cross-entropy loss reduces the mIoU of the baseline S2S (*i.e.* the parsing participating network) by 1.5 from 66.1 when predicting the results one time-step ahead.

Now we proceed to explain the role of the transform layer which transforms the features of CNN$_1$ before combining them with those of CNN$_2$. Compared with naively combining the features from two networks (*e.g.*, concatenation), the transform layer brings the following two advantages: 1) naturally normalize the feature maps to proper scales; 2) align the features of semantic meaning such that the integrated features are more powerful for parsing prediction. Effectiveness of this transform layer is clearly validated in the ablation study in Sec. 4.2.1.

The final objective of our model is to minimize the combination of losses from the flow anticipating network and the parsing anticipating network as follows

$$L(X_{t-k:t-1}, S_{t-k:t-1}, \hat{X}_t, \hat{S}_t) = L_{\text{flow}}(\hat{O}_t, O_t) + L_{\text{seg}}(\hat{S}, S).$$

## 3.2   Prediction for multiple time steps ahead

Based on the above model which predicts scene parsing and flow for the single future time step, we explore two ways to predict further into the future. Firstly, we iteratively apply the model to predict one more time step into the future by treating the prediction as input in a recursive way. Specifically, for predicting multiple time steps in the flow anticipating network, we warp the most recent frame $X_{t-1}$ using the output prediction $\hat{O}_t$ to get the $\hat{X}_t$ which is then combined with $X_{t-k-1:t-1}$ to feed the flow anticipating network to generate $\hat{O}_{t+1}$, and so forth. For the parsing anticipating network, we combine the predicted parsing map $\hat{S}_t$ with $S_{t-k-1:t-1}$ as the input to generate the parsing prediction at $t + 1$. This scheme is easy to implement and allows us to predict arbitrarily far into the future without increasing training complexity w.r.t. with the number of time-steps we want to predict. Secondly, we fine-tune our model by taking into account the influence that the recurrence has on prediction for multiple time steps. We apply our model recurrently as described above to predict two time steps ahead and apply the back propagation through time (BPTT) [14] to update the weight. We have verified through experiments that the fine-tuning approach can further improve the performance as it models longer temporal dynamics during training.

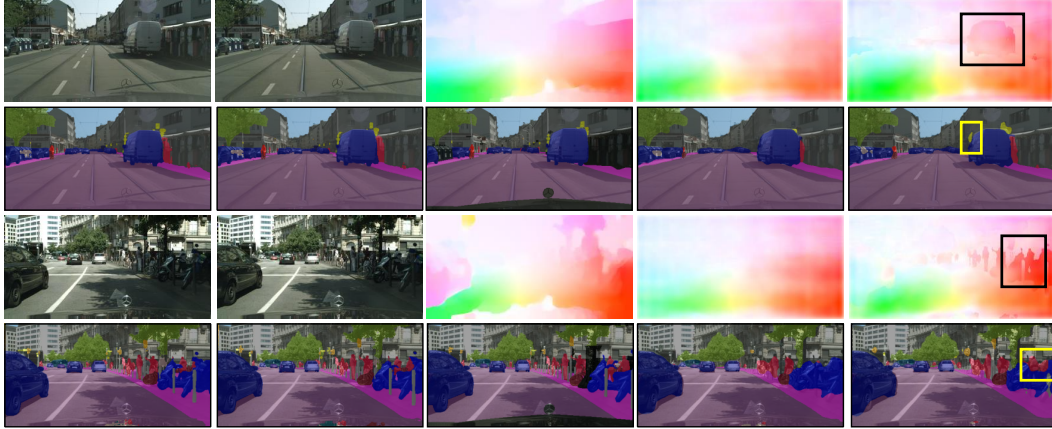

Figure 3: Two examples of prediction results for predicting one time step ahead. **Odd row**: The images from left to right are $X_{t-2}$, $X_{t-1}$, the target optical flow map $O_t$, the flow predictions from PredFlow and the flow predictions from our model. **Even row**: The images from left to right are $S_{t-2}$, $S_{t-1}$, the ground truth semantic annotations at the time $t$, the parsing prediction from $S2S$ and the parsing prediction from our model. The flow predictions from our model show clearer object boundaries and predict more accurate values for moving objects (see black boxes) compared to PredFlow. Our model is superior to S2S by being more discriminative to the small objects in parsing predictions (see yellow boxes).

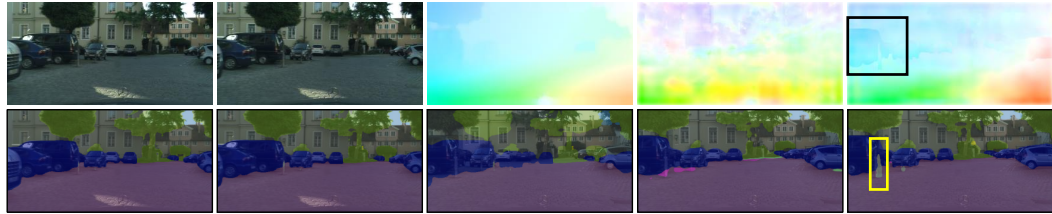

Figure 4: An example of prediction results for predicting ten time steps ahead. **Top** (from left to right): $X_{t-11}$, $X_{t-10}$, the target optical flow map $O_t$, the flow prediction from PredFlow and the flow prediction from our model. **Bottom** (from left to right): $S_{t-11}$, $S_{t-10}$, the ground truth semantic annotation at the time $t$, the parsing prediction from $S2S$ and the parsing prediction from our model. Our model outputs better prediction compared to PredFlow (see black boxes) and S2S (see yellow boxes).

# 4 Experiment

## 4.1 Experimental settings

**Datasets**    We verify our model on the large scale Cityscapes [5] dataset which contains $2,975/500$ train/val video sequences with 19 semantic classes. Each video sequence lasts for $1.8s$ and contains 30 frames, among which the 20th frame has fine human annotations. Every frame in Cityscapes has a resolution of $1,024 \times 2,048$ pixels.

**Evaluation criteria**    We use the mean IoU (mIoU) for evaluating the performance of predicted parsing results on those 500 frames in the val set with human annotations. For evaluating the performance of flow prediction, we use the average endpoint error (EPE) [2] following conventions [8] which is defined as $\frac{1}{N}\sqrt{(u - u_{\text{GT}})^2 + (v - v_{\text{GT}})^2}$ where $N$ is the number of pixels per-frame, and $u$ and $v$ are the components of optical flow along $x$ and $y$ directions, respectively. To be consistent with mIoU, EPEs are also reported on the 20th frame in each val sequence.

**Baselines**    To fully demonstrate the advantages of our model on producing better predictions, we compare our model against the following baseline methods:

Table 1: The performance of parsing prediction on Cityscapes val set. For each competing model, we list the mIoU/EPE when predicting one time step ahead. Best results in bold.

| Model | mIoU | EPE |
|---|---|---|
| Copy last input | 59.7 | 3.03 |
| Warp last input | 61.3 | 3.03 |
| PredFlow | 61.3 | 2.71 |
| S2S [22] | 62.6 | - |
| ours (w/o Trans. layer) | 64.7 | 2.42 |
| ours | **66.1** | **2.30** |

Table 2: The performance of motion prediction on Cityscapes val set. For each model, we list the mIoU/EPE when predicting one time step ahead. Best results in bold.

| Model | mIoU | EPE |
|---|---|---|
| Copy last input | 41.3 | 9.40 |
| Warp last input | 42.0 | 9.40 |
| PredFlow | 43.6 | 8.10 |
| S2S [22] | 50.8 | - |
| ours (w/o Recur. FT) | 52.6 | 6.63 |
| ours | **53.9** | **6.31** |

- *Copy last input*    Copy the last optical flow ($O_{t-1}$) and parsing map ($S_{t-1}$) at time $t-1$ as predictions at time $t$.

- *Warp last input*    Warp the last segmentation $S_{t-1}$ using $O_{t-1}$ to get the parsing prediction at the next time step. In order to make flow applicable to the correct locations, we also warp the flow field using the optical flow in each time step.

- *PredFlow*    Perform flow prediction without the object masks generated from segmentations. The architecture is the same as the flow prediction net in Figure 2 which generates pixel-wise flow prediction in a single layer, instead of multiple branches. For fair comparison with our joint model, in the following we report the average result of two independent *PredFlow* with different random initializations. When predicting the segmentations at time $t$, we use the flow prediction output by *PredFlow* at time $t$ to warp the segmentations at time $t-1$. This baseline aims to verify the advantages brought by parsing prediction when predicting flow.

- *S2S* [22]    Use only parsing anticipating network. The difference is that the former does not leverage features learned by the flow anticipating network to produce parsing predictions. We replace the backbone network in the original S2S as the same one of ours, *i.e.* Res101-FCN and retrain S2S with the same configurations as those of ours. Similar to the *PredFlow*, the average performance of two randomly initialized *S2S* is reported. This baseline aims to verify the advantages brought by flow prediction when predicting parsing.

**Implementation details**    Throughout the experiments, we set the length of the input sequence as 4 frames, *i.e.* $k = 4$ in $X_{t-k:t-1}$ and $S_{t-k:t-1}$ (ref. Sec. 3). The original frames are firstly downsampled to the resolution of $256 \times 512$ to accelerate training. In the flow anticipating network, we assign 19 semantic classes into three object groups which are defined as follows: MOV-OBJ including person, rider, car, truck, bus, train, motorcycle and bicycle, STA-OBJ including road, sidewalk, sky, pole, traffic light and traffic sign and OTH-OBJ including building, wall, fence, terrain and vegetation. For data augmentation, we randomly crop a patch with the size of $256 \times 256$ and perform random mirror for all networks. All results of our model are based on single-model single-scale testing. For other hyperparameters including weight decay, learning rate, batch size and epoch number *etc*., please refer to the supplementary material. All of our experiments are carried out on NVIDIA Titan X GPUs using the Caffe library.

## 4.2   Results and analysis

Examples of the flow predictions and parsing predictions output by our model for one-time step and ten-time steps are illustrated in Figure 3 and Figure 4 respectively. Compared to baseline models, our model produces more visually convincing prediction results.

### 4.2.1   One-time step anticipation

Table 1 lists the performance of parsing and flow prediction on the 20th frame in the val set which has ground truth semantic annotations. It can be observed that our model achieves the best performance on both tasks, demonstrating the effectiveness on learning the latent representations for future prediction. Based on the results, we analyze the effect of each component in our model as follows.

**The effect of flow prediction on parsing prediction** Compared with S2S which does not leverage flow predictions, our model improves the mIoU with a large margin (3.5%). As shown in Figure 3, compared to S2S, our model performs better on localizing the small objects in the predictions *e.g.* pedestrian and traffic sign, because it combines the discriminative local features learned in the flow anticipating network. These results clearly demonstrate the benefit of flow prediction for parsing prediction.

**The effect of parsing prediction on flow prediction** Compared with the baseline PredFlow which has no access to the semantic information when predicting the flow, our model reduces the average EPE from 2.71 to 2.30 (a 15% improvement), which demonstrates parsing prediction is beneficial to flow prediction. As illustrated in Figure 3, the improvement our model makes upon PredFlow comes from two aspects. First, since the segmentations provide boundary information of objects, the flow map predicted by our model has clearer object boundaries while the flow map predicted by PredFlow is mostly blurry. Second, our model shows more accurate flow predictions on the moving objects (ref. Sec. 4.1 for the list of moving objects). We calculate the average EPE for only the moving objects, which is 2.45 for our model and 3.06 for PredFlow. By modeling the motion of different objects separately, our model learns better representation for each motion mode. If all motions are predicted in one layer as in PredFlow, then the moving objects which have large displacement than other regions are prone to smoothness.

**Benefits of the transform layer** As introduced in Sec. 3.1, the transform layer improves the performance of our model by learning the latent feature space transformations from $CNN_1$ to $CNN_2$. In our experiments, the transform layer contains one residual block [12] which has been widely used due to its good performance and easy optimization. Details of the residual block used in our experiments are included in the supplementary material. Compared to the variant of our model w/o the transform layer, adding the transform layer improves the mIoU by 1.4 and reduces EPE by 0.12. We observe that stacking more residual blocks only leads to marginal improvements at larger computational costs.

### 4.2.2 Longer duration prediction

The comparison of the prediction performance among all methods for ten time steps ahead is listed in Table 2, from which one can observe that our model performs the best in this challenging task. The effect of each component in our model is also verified in this experiment. Specifically, compared with S2S, our model improves the mIoU by 3.1% due to the synergy with the flow anticipating network. The parsing prediction helps reducing the EPE of PredFlow by 1.79. Qualitative results are illustrated in Figure 4.

**The effect of recurrent fine-tuning** As explained in Sec. 3.2, it helps our model to capture long term video dynamics by fine-tuning the weights when recurrently applying the model to predict the next time step in the future. As shown in Table 2, compared to the variant w/o recurrent ft, our model w/ recurrent fine-tuning improves the mIoU by 1.3% and reduces the EPE by 0.32, therefore verifying the effect of recurrent fine-tuning.

### 4.3 Application for predicting the steering angle of a vehicle

With the parsing prediction and flow prediction available, one can enable the moving agent to be more alert about the environments and get "smarter". Here, we investigate one application: predicting the steering angle of the vehicle. The intuition is it is convenient to infer the steering angle given the predicted flow of static objects, *e.g.* road and sky, the motion of which is only caused by ego-motion of the camera mounted on the vehicle. Specifically, we append a fully connected layer to take the features learned in the STA-OBJ branch in the flow anticipating network as input and perform regression to steering angles. We test our model on the dataset from Comma.ai [27] which consists of 11 videos

Table 3: Comparison results of steering angle prediction on a dataset from Comma.ai [27]. The criteria is the mean square error (MSE, in $degree^2$) between the prediction and groud truth.

| Model | MSE ($degrees^2$) |
|---|---|
| Copy last prediction | 4.81 |
| Comma.ai[1] [27] | $\sim 4$ |
| ours | 2.96 |

amounting to about 7 hours. The data of steering angles have been recorded for each frame captured at 20Hz with the resolution of $160 \times 320$. We randomly sample 50K/5K frames from the train set for training and validation purpose. Since there are videos captured at night, we normalize all training frames to $[0, 255]$. Similar to Cityscapes, we use epicflow and Res101-FCN to produce the target output for flow prediction and parsing prediction, respectively. We first train our model following Sec. 3 and then fine-tune the whole model with the MSE loss after adding the fully connected layer for steering angle prediction. During training, random crop with the size of $160 \times 160$ and random mirror are employed and other hyperparameter settings follow Sec. 4.1. The testing results are listed in Table 3. Compared to the model from Comma.ai which uses a five-layer CNN to estimate the steering angle from a single frame and is trained end-to-end on all the training frames (396K), our model achieves much better performance (2.84 versus $\sim$4 in $degrees^2$). Although we do not push the performance by using more training data and more complex prediction models (only a fully connected layer is used in our model for output steering angle), this preliminary experiment still verifies the advantage of our model in learning the underlying latent parameters. We think it is just an initial attempt in validating the dense prediction results through applications, which hopefully can stimulate researchers to explore other interesting ways to utilize the parsing prediction and flow prediction.

## 5  Conclusion

In this paper, we proposed a novel model to predict the future scene parsing and motion dynamics. To our best knowledge, this is the first research attempt to anticipate visual dynamics for building intelligent agents. The model consists of two networks: the flow anticipating network and the parsing anticipating network which are jointly trained and benefit each other. On the large scale Cityscapes dataset, the experimental results demonstrate that the proposed model generates more accurate prediction than well-established baselines both on single time step prediction and multiple time prediction. In addition, we also presented a method to predict the steering angle of a vehicle using our model and achieve promising preliminary results on the task.

**Acknowledgements**   The work of Jiashi Feng was partially supported by National University of Singapore startup grant R-263-000-C08-133, Ministry of Education of Singapore AcRF Tier One grant R-263-000-C21-112 and NUS IDS grant R-263-000-C67-646.

## Footnotes

[1] https://github.com/commaai/research

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
