[Supplementary Material · supplementary_material.pdf]

# Predicting Scene Parsing and Motion Dynamics in the Future (Supplementary Material)

**Xiaojie Jin**[1]**, Huaxin Xiao**[2]**, Xiaohui Shen**[3]**, Jimei Yang**[3]**, Zhe Lin**[3]
**Yunpeng Chen**[2]**, Zequn Jie**[4]**, Jiashi Feng**[2]**, Shuicheng Yan**[5,2]
[1]NUS Graduate School for Integrative Science and Engineering (NGS), NUS
[2]Department of ECE, NUS    [3]Adobe Research    [4]Tencent AI Lab    [5]Qihoo 360 AI Institute

## Abstract

In supplementary material, we provide the architecture details and training settings of Res101-FCN and the transform layer.

## 1 Implementation Details

### 1.1 Network Architecture

The architecture of our Res101-FCN is illustrated in Figure 1. It is modified from the original Res101 [2] by adapting it to a fully convolutional network, following [3]. Specifically, we replace the average pooling layer and the 1,000-way classification layer with a fully convolutional layer (denoted as $\text{conv5\_3}_{\text{cls}}$ in Figure 1) to produce dense parsing maps. Also, we modify `conv5_1`, `conv5_2` and `conv5_3` to dilated convolutional layers by setting their dilation size to 2 to enlarge the size of the receptive field. Then, the output feature map of `conv5_3` has a stride of 16. Following FCN [3], we utilize high-frequency features learned in bottom layers by adding skip connections from `conv1`, `pool1`, `conv3_3` to corresponding up-sampling to produce parsing maps with the same size as input frames. The training setting are listed at Table 1. The mIoU of Res101-FCN on val of Cityscapes is 75.2%.

### 1.2 Architecture of Transform Layer

In our experiments, a "bottleneck" residual block proposed in [2] is used as the transform layer whose architecture is illustrated in Figure 2.

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

Figure 1: Architecture of Res101-FCN used in our experiments. The subscript "cls" means the corresponding layer is a convolutional layer which has the same number of filters as the categorical classes, *e.g.* the `conv5_3`$_{cls}$ that has 19 filters for Cityscapes dataset. Following FCN [3], skip connections from `conv1`, `pool1`, `conv3_3` are constructed to utilize high frequency information in bottom layers. The `up`$_n$ is an up-sampling layer, of which the output feature map is $n$ times the spatial size of that of `conv5_3`$_{cls}$. The symbol $\oplus$ represents the operation of "summation".

Figure 2: Architecture of the residual block [2] used as the transform layer in our model. In the residual block, the shortcut connection performs identity mapping, the output of which is added to the output of the stacked convolutional layers.

Table 1: Training settings of Res101-FCN. "LR" stands for "learning rate". In all experiments, the weight decay and momentum are set to 0.0001 and 0.9 respectively. Following Deeplab [1], the "poly" learning rate policy is used and the value of power is equal to 0.9.

| Batch size | Crop size | Overall epochs | LR-initial | LR-policy |
|---|---|---|---|---|
| 4 | 256 | 30 | 1e-4 | Poly |