[Reviews · NeurIPS 2017]

Reviewer 1



The paper proposes a deep-learning-based approach to joint prediction of future optical flow and semantic segmentation in videos. The authors evaluate the approach in a driving scenario and show that the two components - flow prediction and semantic segmentation prediction - benefit from each other. moreover, they show a proof-of-concept experiment on using the predictions for steering angle prediction. The paper is related to works of Jin et al. and Neverova et al. However, as far as I understand, both of these have not been officially published at the time of submission (and the work of Neverova et al. only appeared on arxiv 2 months before the deadline), so they should not be considered prior art. Detailed comment: Pros: 1) The idea seems sound: predicting segmentation and optical flow are both important tasks, and they should be mutually beneficial. 2) The architecture is clean and seems reasonable 3) Experimental evaluation on Citiscapes is reasonable, and some relevant baselines are evaluated: copying previous frame, warping with the estimated flow, predicting segmentation without flow, predicting flow without segmentation, the proposed model without the “transformation layer”. The improvements brought by the modifications are not always huge, but are at least noticeable. Cons: 1) Steering angle prediction experiments are not convincing at all. The baseline is very unclear, and it is unclear what does the performance of “~4” mean. There is no obvious baseline - training a network with the same architecture as used by the authors, but trained from scratch. The idea of the section is good, but currently it’s not satisfactory at all. 2) I am confused by the loss function design. First, the weighting of the loss component is not mentioned at all - is there indeed no weighting? Second, the semantic segmentation loss is different for human-annotated and non-human-annotated frames - why is that? I understand why human-annotated loss should have a higher weight, but why use a different function - L1+GDL instead of cross-entropy? Have you experimented with other variants? 3) Figures 3 and 4 are not particularly enlightening. Honestly, they do not tell too much. Of course it is always possible to find a sample on which the proposed method works better. The authors could find a better way to represent the results - perhaps with a video, or with better-designed figures. 4) Writing looks rushed and unpolished. For instance, caption of Table 2 is wrong. Minor: 5) I’m somewhat surprised the authors use EpicFlow - it’s not quite state of the art nowadays 6) Missing relevant citation: M. Bai, W. Luo, K. Kundu and R. Urtasun: Exploiting Semantic Information and Deep Matching for Optical Flow. ECCV 2016. 7) Many officially published papers are cited as arxiv, to mention a few: Cityscapes, FlowNet, ResNet, Deep multi-scale video prediction, Context encoders Overall, the paper is interesting and novel, but seems somewhat rushed, especially in some sections. Still, I believe it could be accepted if the authors fix some most embarrassing problems mentioned above, in particular, the part on steering angle prediction.

Reviewer 2



This paper combines two ideas: 1) predicting optical flow and 2) image segmentation, both using convolutional architectures on a multi-step context, into a joint architecture that both predicts optical flow using semantic labels and predicts the semantic labels S_{t+1} on the next frame using previous optical flow. The paper thereby introduces the idea of predicting the semantic label on the next image rather than on the current image. Specifically, the flow predictor outputs one flow map for foreground moving objects, one for static backgrounds and one for other (e.g., vegetation), and the pixel assignment to these 3 classes depends on the semantic segmentation. The flow features (before upsampling and up-convolution) are used as extra inputs, through a transform layer, along with previous semantic maps, to predict the future semantic map. Finally, the k-step context CNN can be used as an auto-regressive model on S_t and flow to generate iterated predictions, and be trained like an unrolled RNN, using BPTT. The model is trained on automatically generated labels, provided by state-of-the-art semantic image segmenters and flow estimators. The results of the joint model outperform both epicflow [21] and S2S [18]. The application to steering angle prediction uses a single baseline. As this seems not to be a common research problem in the community, these experimental results seem anecdotical and would be enhanced by comparisons to more methods. The paper could benefit from some rewriting. The introduction is very verbose, and that space could be used for a more detailed figure of the full architecture. Are S_{t-1}, ..., S_{t-4} produced by the same epicflow model that is used for distillation, or is the flow predictor retrained? The abstract also contains grammatical errors, e.g., "later" -> "latter", "arbitrary time steps ahead".

Reviewer 3



The paper introduces a joint system for predicting motion flow and future scene segmentations from frames of video. The motivation is that flow and segmentation are co-dependent problems, and might mutually benefit from each others' solutions. The authors use two CNN architectures, one which targets motion flow prediction and the other which targets future scene parsing, with a mid-level connection from the motion predictor to the scene parser. They are trained jointly. Their results show their trained system outperforms PredFlow on flow prediction and outperforms S2S on future scene parsing. In general the paper is interesting but has several problems. 1. The writing and presentation are not very clear in many place. There was little intuition for why this should or shouldn't work. I had an especially hard time understanding details of the proposed architecture, such as what the inputs are. For example, how exactly does the flow network benefit from the parsing network? Is it just through their joint training? The Intro says: "In the motion prediction task, we utilize the semantic segmentations produced by predictive parsing to separately predict motion for pixels with different categories." And section 3.1 says: "First, the last segmentations St−1 produced by the parsing anticipating network convey pixel-wise class labels, using which the flow anticipating network can predict optical flow values for each pixel according to its belonging object group, e.g. moving objects or static objects." However, I do not see this made explicit anywhere. How does this work? 2. The novelty is relatively small given previous work. The paper focuses on joining two strong approaches together and showing that modest improvements are possible in their respective tasks. The improvement in anticipatory scene parsing isn't that surprising - if the motion prediction features weren't useful, the scene parser should just learn to ignore them, and of course motion flow is very sensitive to object boundaries. I'm less sure I understand the reason for the improvement in motion prediction due to my point above about not understanding how the motion predictor uses the parser's output. But again, because both tasks depend on common underlying spatial structure in scenes, it isn't too surprising that parsing features can improve motion prediction. 3. Isn't this an example of multi-task learning, or at least a close variant? It would be nice to acknowledge that literature and situate this work in the context of what's done there. One suggestion for making this work more relevant to those who don't necessarily work in this specific visual domain would be to further develop this as a general purpose framework for joining architectures that specialize in different tasks, so a user can take two good architectures and have a good chance of their hybrid learning both tasks better. I think this work moves in that direction, but I don't think it explores this sufficiently (e.g. the transform layer).